# An integrated blockchain and IPFS-based solution for secure and efficient source code repository hosting using middleman approach

**Md. Rafid Haque[1], Sakibul Islam Munna[1], Sabbir Ahmed** **[1]\*, Md. Tahmid Islam[1], Md Mehedi Hassan Onik[1,2], A.B.M. Ashikur Rahman[1,3]**

**1** Department of Computer Science and Engineering, Islamic University of Technology (IUT), Boardbazar, Gazipur, Bangladesh, **2** School of IT, Deakin University, Waurn Ponds, Victoria, Australia, **3** Department of ICS, King Fahd University of Petroleum & Minerals, Dhahran, Saudi Arabia

\* sabbirahmed@iut-dhaka.edu

## Abstract

Centralized version control systems (VCS) are vital for software development but pose risks of data loss and ownership disputes. While blockchain offers a decentralized alternative, existing solutions are often hindered by high latency, compromising the real-time collaboration essential for modern workflows. This study introduces a novel hybrid architecture combining the security of the Ethereum blockchain and the InterPlanetary File System (IPFS) with two key contributions: 1) Shamir's Secret Sharing (SSS) to create a trust-minimized model for key distribution, and 2) an authoritative-first, optimistic-fallback retrieval protocol utilizing a temporary middleware to decouple the user experience from blockchain confirmation delays. We implemented a full prototype and conducted a comprehensive performance evaluation on the public Sepolia testnet. Our results demonstrate that this architecture not only provides a secure, auditable, and resilient platform for source code hosting but also achieves highly competitive user-perceived performance. Our user-perceived push time reduces submission latency by up to 49% compared to a standard git push for common repository sizes, proving that a well-designed decentralized VCS can balance the core tenets of security and decentralization with the practical need for speed and efficiency.

## Introduction

Software development projects often rely on version control systems (VCS) to track and manage their code and files. However, most existing VCS are centralized, which means that they depend on a single authority or service provider that can pose risks of data loss, security breaches, and ownership disputes [1–3]. Therefore, there is a need for a decentralized, reliable, and secure solution for code repository hosting and governance. Blockchain technology is a promising candidate for such a solution, as it enables a distributed ledger that is immutable, transparent, and consensus-based, without any trusted intermediaries [4,5].

**Data availability statement:** All relevant data for this study are within the paper, its Supporting Information files, and publicly

available from the Zenodo repository
(https://doi.org/10.5281/zenodo.15700465).

**Funding:** The author(s) received no specific funding for this work.

**Competing interests:** The authors have declared that no competing interests exist.

The use of self-executing smart contracts can further automate processes, reduce the need for intermediaries, and enforce predefined rules for digital transactions with high security [6–12].

While blockchain has been successfully applied to enhance trust and transparency in domains like supply chain management [13–17] and healthcare [18–23], its application to performance-sensitive domains like version control faces a critical obstacle. The primary challenge is the inherent latency of on-chain transactions. A system requiring every action to await a blockchain confirmation, which can take several seconds or even minutes, creates a poor user experience that hinders real-time collaboration and prevents widespread adoption [24–26]. This performance bottleneck has been a significant barrier to the development of practical, decentralized software development tools.

To address this critical latency challenge, we look to established architectural patterns from the literature. Research has shown that using an off-chain "fast path" or cache is a recognized strategy for enhancing the performance of blockchain systems [27–29]. Similarly, the use of cryptographic techniques like Shamir's Secret Sharing (SSS) has been validated in other high-security domains to distribute trust and eliminate single points of failure [30–34]. Building upon these validated concepts, we propose a novel hybrid architecture that makes two core contributions:

1. We introduce a trust-minimized security model specifically for VCS that uses SSS to distribute the cryptographic repository key.
2. We design an authoritative-first, optimistic-fallback protocol that uses a lightweight middleware to make the user workflow highly responsive, effectively hiding the blockchain's transaction delay from the end-user.

We have implemented this system as a full-stack decentralized application and conducted a comprehensive performance evaluation on the public Sepolia testnet. Our results demonstrate that this architecture not only provides a secure and auditable platform but also achieves highly competitive performance. Most notably, we show that our system's user-perceived push time can be faster than a standard 'git push' for common repository sizes, proving that a carefully designed decentralized VCS can balance robust security with the practical need for speed and efficiency. This paper is organized as follows. The Related works section reviews the literature on decentralized version control and relevant architectural patterns. The Materials and methods section details our proposed system architecture and protocols. The Results and discussion section presents our experimental results and a detailed performance analysis. Finally, the Conclusion section concludes the paper and discusses future work.

## Related works

This section reviews the literature across three key domains that inform our work: (1) existing decentralized version control systems and their limitations; (2) architectural patterns for managing decentralized trust and security; and (3) strategies for mitigating the performance latency of blockchain systems.

### Decentralized version control: Prior foundational challenges

Several early works have explored the application of blockchain technology to version control systems (VCS). One of the first, Capivara, proposed a decentralized package version control system using a proof-of-download consensus approach [35]. While conceptually innovative, the work did not include an implementation or empirical evaluation, leaving its practical

effectiveness unexplored. Another foundational study by Nizamuddin et al. used Ethereum smart contracts and the InterPlanetary File System (IPFS) for document version control [36]. The authors demonstrated the efficiency of using IPFS for decentralized storage; however, their reliance on synchronous Ethereum transactions for every version control action highlighted the significant time delays that could be prohibitive in real-world, collaborative scenarios.

Subsequent research has built upon these foundations. Systems like BDA-SCV by Hammad et al. [37] and PineSU by Grilli and Speziali [38] have created functional systems that directly combine Git workflows with a blockchain backend to ensure data integrity and authenticity. Other works have focused on related use cases, such as providing auditable versioning for scientific and educational artifacts [39–41]. While these systems advance the field, a recurring theme is the performance trade-off, where the added security of on-chain operations often results in increased latency, underscoring the critical need for a solution that prioritizes user-perceived speed [42,43].

Other works have focused on using blockchain for intellectual property (IP) management. The application of smart contracts to protect digital music [44,45] and literary IP [46–49] showcases the potential for blockchain to provide a transparent and tamper-resistant approach to managing ownership records. However, these systems, like RecordsKeeper [50] and the solution by Eleks Labs [51], typically focus on authenticity and rights management rather than the specific, high-frequency, collaborative workflows required by a VCS. A systematic mapping study by Demi et al. on blockchain in software engineering confirmed the technology's potential but also emphasized the significant challenges of scalability and complexity that must be overcome for practical adoption [52].

## Decentralized trust and key management via secret sharing

A fundamental challenge in decentralized systems is managing authority and access to shared resources without a central administrator. The literature has increasingly converged on secret sharing schemes as a powerful cryptographic primitive for distributing trust. These schemes allow a secret key, such as a master encryption key, to be split into multiple shares, requiring a threshold of participants to collaborate to reconstruct it. This approach provides a robust defense against single points of failure and malicious attacks, a principle explored in contexts ranging from generic cloud databases [34,53] to the protection of high-value crypto assets [54].

This architectural pattern has been explicitly validated in various high-stakes domains that require both high security and data integrity. In the context of the Internet of Things (IoT) and vehicular ad-hoc networks (VANETs), where ensuring trust is pivotal, combining blockchain with a secret sharing mechanism has become a state-of-the-art solution. Works by Kim et al. [32] and Mao [55] demonstrate the use of Shamir's Secret Sharing (SSS) and blockchain to secure document management and sensitive medical data on IPFS . This principle is further extended by Bansal et al. [56] and Nakkar et al. [57], who leverage SSS to build lightweight and reliable authentication schemes for resource-constrained devices like Unmanned Aerial Vehicles (UAVs).

Furthermore, a significant body of work focuses on building comprehensive, decentralized trust management frameworks. The work by Razzaq et al. is notable in this area, establishing clear precedents for using blockchain as an immutable trust anchor to coordinate interactions and manage data securely in diverse applications such as educational platforms, healthcare, and vehicular networks [58–65]. Similarly, studies on VANETs by Gazdar et al. [66],

Chen et al. [67], and Pu et al. [68] propose blockchain-based systems to verify the credibility of messages and manage trust between vehicles. These works, along with comprehensive surveys on blockchain for cybersecurity [69], confirm that combining a blockchain ledger for auditability with cryptographic techniques for distributed trust is a state-of-the-art methodology.

### Architectural patterns for mitigating blockchain latency

While cryptographic mechanisms can solve for decentralized trust, they do not inherently address the performance limitations of the underlying blockchain ledger. The literature has extensively explored this challenge, converging on a primary strategy: moving the bulk of storage and computation off-chain, while using the blockchain itself as a lightweight anchor for verification and asynchronous settlement. This "off-chain first" philosophy is critical for making decentralized applications practical and responsive.

Amongst several works validating this approach, systems like Saguaro by Amiri et al. and FLCoin by Ren et al. use hierarchical or layered blockchain architectures to reduce communication overhead and consensus latency in edge computing environments [70, 71]. Others focus on optimizing the consensus layer itself, with protocols like Banyan and Remora introducing "fast paths" or "optimistic paths" to reduce block finalization time [72,73]. A particularly relevant strategy is the use of off-chain caching. Kim and Park, for instance, propose a distributed caching architecture to guarantee real-time responsiveness for DApps [74,75], while Liang et al. introduce inter-shard caching in their Sparrow protocol to expedite smart contract execution [76]. This same principle of improving performance by separating concerns is also seen in the domain of Network Function Virtualization (NFV), where service chains are orchestrated to balance latency and resource costs [24,77–79].

Our system applies this validated "off-chain fast path" pattern directly to the version control workflow. The middleman component in our architecture functions as a specialized, temporary cache for key shares, analogous to the off-chain layers in the aforementioned systems. This design choice allows us to optimize for user-perceived latency, ensuring that developer workflows are not blocked by on-chain confirmation times. Our asynchronous design achieves a highly responsive user experience, a significant advancement for usable decentralized applications.

## Materials and methods

In this section, we describe our proposed method, which uses the Ethereum blockchain, the InterPlanetary File System (IPFS), and a hybrid cryptographic approach to authorize, monitor, and maintain version control for code repositories. A key innovation of our system is the elimination of a single point of trust in the key management process through the use of Shamir's Secret Sharing (SSS), which allows us to distribute trust between the blockchain and a temporary centralized middleware. This architecture allows for the secure sharing and tracking of different versions of code while mitigating the inherent latency of public blockchains.

### Push process

The repository **push process**, as illustrated in Fig 1, is designed to be asynchronous to optimize the user experience. It begins with the client-side encryption of the user's source code. The encrypted repository is uploaded to a decentralized storage provider, generating a unique

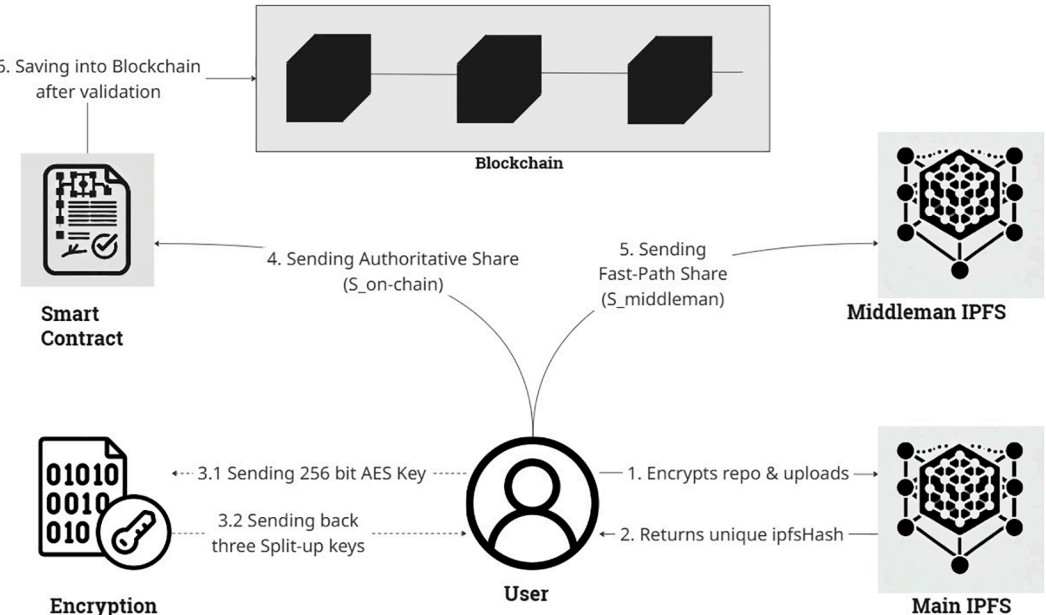

**Fig 1. Proposed system push process: Source code is encrypted and stored on IPFS.** The encryption key is split into three shares via SSS: One is retained by the owner, one is sent to the middleman for fast retrieval, and one is registered on the blockchain asynchronously.

IPFS Content Identifier (CID). To secure the repository's encryption key, we employ a (k,n)-threshold secret sharing scheme. The resulting key shares are strategically distributed: one share is sent to a temporary middleman for optimistic fallback retrieval; another is registered on the Ethereum blockchain as the authoritative, long-term record in a background transaction; and the final share is retained by the repository owner.

- **Client-Side Encryption and IPFS Upload**: The process initiates on the user's client machine. The source code repository (e.g., a .zip file) is encrypted using a newly generated 256-bit AES symmetric key. This encrypted data blob is then uploaded to a decentralized storage service compatible with IPFS, such as Pinata, which returns a unique and immutable IPFS CID. This CID serves as the permanent address for the encrypted repository.
- **Key-Share Generation via Shamir's Secret Sharing (SSS)**: To address the trust and latency issues of storing a complete key, we enhance our security model with SSS. The 256-bit AES key (bundled with its Initialization Vector) is treated as a single secret. This secret key is then split into $n$ unique shares using a $(k,n)$-threshold scheme (e.g., $k = 2, n = 3$). In this scheme, any $k$ shares can reconstruct the original secret, but any $k-1$ shares reveal no information, cryptographically eliminating a single point of failure.
- **Decentralized Share Distribution**: The generated key shares are distributed to distinct entities to ensure resilience and facilitate our optimistic retrieval protocol.
  - **Owner's Share:** One share is immediately returned to the repository owner. This share acts as the primary "key" that the owner can give to collaborators to grant access.

- **Middleman's Share:** A second share is sent to a lightweight, temporary middleman middleware. This share is cached and made available for immediate retrieval, serving as the fast fallback path of our protocol.
- **On-Chain Share:** The third share is included in a transaction sent to our smart contract on the Ethereum blockchain. This serves as the authoritative, primary path—the immutable, long-term record that is queried first during retrieval.

## Pull process

The repository **pull process**, as shown in Fig 2, employs an authoritative-first, optimistic-fallback model. The first step is always a non-blocking call to the smart contract to verify the collaborator's permission. Once access is granted, the system prioritizes reliability by first attempting to fetch the required key share from the authoritative on-chain source. Only if this primary path is unavailable (e.g., because the submission transaction is still pending confirmation) does the system automatically fall back to the fast middleman path. This design ensures that the most trustworthy data source is always preferred, while the middleman provides a crucial mechanism to bypass latency and ensure a responsive user experience.

- **Permission Verification**: Before any data retrieval, the application makes a view call to the smart contract's 'checkAccess' function. This provides an immediate, authoritative check of the user's permissions. The process only continues if access is granted.
- **Authoritative Key-Share Retrieval**: The client application first calls the 'getOnChainShare' function of the smart contract. If the transaction has been confirmed and the share is present, it is returned, and the system proceeds directly to key reconstruction.

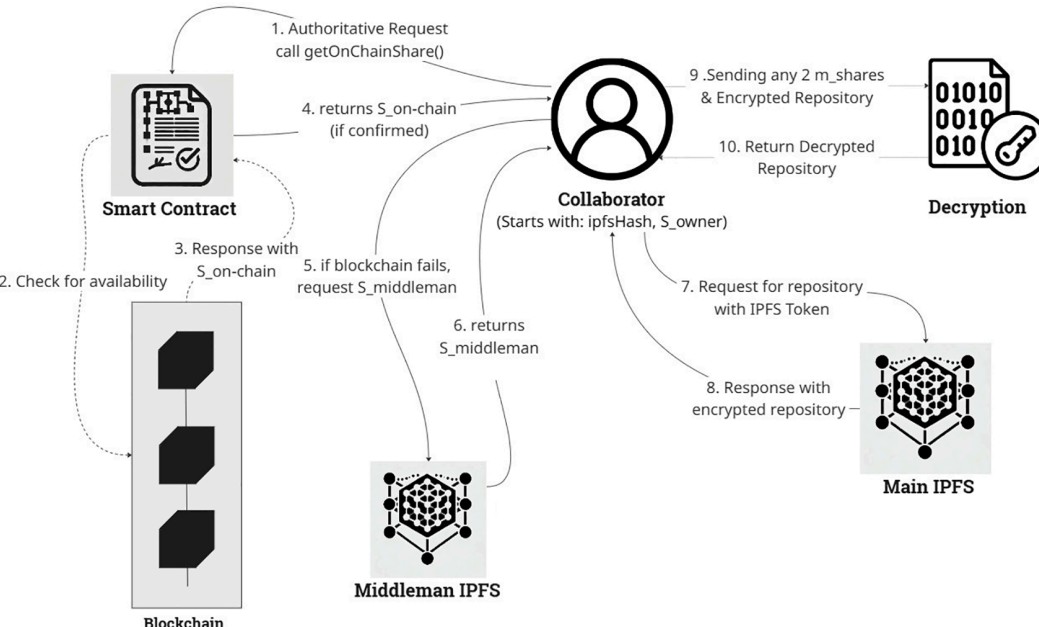

**Fig 2. Proposed system pull process: After an on-chain permission check, the system first queries the authoritative blockchain for a key share.** If unavailable due to latency, it optimistically falls back to the middleman, ensuring a responsive user experience.

- **Optimistic Fallback Retrieval**: If the on-chain call fails or returns an empty value (indicating a pending transaction), the application automatically falls back to the "optimistic path." It sends a request to the middleman middleware to fetch the second required share.
- **Key Reconstruction and Decryption**: Once any two shares are successfully retrieved from either source, the client-side SSS algorithm combines them to perfectly reconstruct the original AES key and IV. The application then uses the IPFS CID to fetch the encrypted data blob from a public IPFS gateway (e.g., 'gateway.pinata.cloud'). The downloaded content is decrypted in-memory using the reconstructed key, yielding the original source code.

## Environment setup and implementation

The system was developed as a client-side web application using standard web technologies (HTML, CSS, and JavaScript), leveraging the `Ethers.js` (https://docs.ethers.org/v5/) library for blockchain interaction. For decentralized storage, we integrated with the Pinata (https://www.pinata.cloud/) pinning service for uploads and a public IPFS gateway for downloads. To implement our optimistic retrieval protocol, a lightweight middleman server was developed using Node.js (https://nodejs.org/) and Express (https://expressjs.com/), and deployed as a serverless function. For testing purposes, we first used a local blockchain environment (Ganache; https://trufflesuite.com/ganache/) to validate functionality. Subsequently, the system was deployed to the public Sepolia Testnet (https://sepolia.etherscan.io/) to conduct a comprehensive performance evaluation in a real-world setting.

The development environment included the Remix IDE (https://remix.ethereum.org/) for writing and deploying our Solidity smart contract. Two Ethereum addresses were used for testing: one representing the repository owner and another for the collaborator. Each participant was provided with test Ether on the Sepolia network to facilitate transactions and validate the correctness of our cryptographic and access control mechanisms.

The implementation leverages client-side JavaScript to perform all cryptographic operations and interactions with the blockchain and middleware, ensuring that private keys and unencrypted data never leave the user's machine. The entire experimental setup was designed to rigorously test the performance trade-offs between security, decentralization, and the operational efficiency required for real-time collaboration.

**Code availability.** The source code for the client-side dApp, the Node.js middleman server, and the Solidity smart contract developed for this study is publicly available in a GitHub repository [80] (https://github.com/rafidhaque/Blockchain-and-middleman-ipfs-based-solution-for-repository-hosting).

## Results and discussion

In this section, we present the empirical results from a comprehensive performance evaluation of our proposed system. The experiments were designed to quantify the user-perceived latency of core developer workflows, identify the underlying system bottlenecks, and provide a direct comparison against a centralized baseline (Git/GitHub). Our findings demonstrate that by architecturally decoupling the user's workflow from the inherent latency of public

**Algorithm 1. Pseudocode for the smart contract.**

```
1: Initialize State Variables:
2:    Mapping owners: string (ipfsHash) → address
3:    Mapping hasAccess: string (ipfsHash) → (address → bool)
4:    private Mapping onChainShares: string (ipfsHash) → string (keyShare)

5: procedure RegisterRepository(ipfsHash, onChainShare)
6:    Require: owners[ipfsHash] is not set
7:    owners[ipfsHash] ← msg.sender
8:    hasAccess[ipfsHash][msg.sender] ← true
9:    onChainShares[ipfsHash] ← onChainShare
10: end procedure

11: function GetOnChainShare(ipfsHash)
12:    Require: hasAccess[ipfsHash][msg.sender] is true
13:    Return: onChainShares[ipfsHash]
14: end function

15: function CheckAccess(ipfsHash, userAddress)
16:    Return: hasAccess[ipfsHash][userAddress]
17: end function

18: procedure AddCollaborator(ipfsHash, collaboratorAddress)
19:    Require: msg.sender is owners[ipfsHash]
20:    hasAccess[ipfsHash][collaboratorAddress] ← true
21: end procedure
```

**Algorithm 2. Cryptographic and distribution process.**

1: **Input:** Source Code Repository R
2: **Output:** Owner's Share $S_{\text{owner}}$, IPFS CID $H_{\text{ipfs}}$

3: **procedure** SubmitRepository(R)
4:    **Step 1: Client-Side Encryption**
5:    Generate a symmetric key $K_{\text{sym}}$ and initialization vector IV
6:    $R_{\text{enc}} \leftarrow \text{Encrypt}_{\text{AES}}(R, K_{\text{sym}}, IV)$
7:
8:    **Step 2: Decentralized Storage**
9:    $H_{\text{ipfs}} \leftarrow \text{UploadToIPFS}(R_{\text{enc}})$
10:
11:    **Step 3: Key Sharing and Distribution**
12:    $Secret \leftarrow \text{Concatenate}(K_{\text{sym}}, IV)$
13:    $\{S_{\text{owner}}, S_{\text{on-chain}}, S_{\text{middleman}}\} \leftarrow \text{Split}_{\text{SSS}}(Secret, k=2, n=3)$
14:    SendToMiddlemanServer($H_{\text{ipfs}}, S_{\text{middleman}}$)                                    ▷ Fast Path
15:    Call smart contract: RegisterRepository($H_{\text{ipfs}}, S_{\text{on-chain}}$)     ▷ Authoritative Path
16:    **Return** $S_{\text{owner}}$, $H_{\text{ipfs}}$
17: **end procedure**

blockchains, our hybrid system not only provides the security benefits of decentralization but also achieves highly competitive, and in some cases, superior performance.

## Performance evaluation

To quantitatively assess our system, we conducted a series of experiments on the Sepolia test-net. We defined "push latency" from a user-experience perspective: the time until a repository is available for retrieval by a collaborator. In our system, this occurs immediately after the file is uploaded to IPFS and its corresponding key share is stored in the middleman, without waiting for blockchain confirmation. All tests were repeated five times across various file sizes (1, 5, 10, and 20 MB).

**Comparative analysis with centralized VCS.** The primary goal of our evaluation was to contextualize our system's performance against the industry standard. Table 1 summarizes the average user-perceived latency for core operations, with Fig 3 visualizing the comparison.

The results are striking. For repositories up to 10MB, the user-perceived push time of our proposed system is faster than a standard 'git push'. This is achieved by architecturally treating the time-consuming blockchain transaction as an asynchronous background process, which allows the developer to continue their workflow without interruption. While the performance

**Table 1. User-perceived performance: proposed system vs. centralized git (in seconds).**

| Size | System Push (s) | Git Push (s) | System Pull (s) | Git Pull (s) |
|---|---|---|---|---|
| 1 MB | 2.04 | 4.05 | 1.29 | 1.06 |
| 5 MB | 4.19 | 6.16 | 2.41 | 1.07 |
| 10 MB | 6.56 | 7.74 | 2.54 | 1.06 |
| 20 MB | 11.47 | 8.14 | 4.08 | 1.17 |

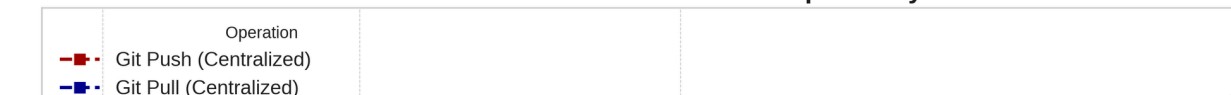

### User-Perceived Performance: Decentralized Proposed System vs. Centralized Git

**Fig 3. Performance comparison: proposed system vs. centralized git.** User-perceived push time is faster than git push for common repository sizes, while pull performance is highly competitive, validating our asynchronous, optimistic design.

for very large files (20MB) is eventually limited by the IPFS upload speed, the overall submission performance is highly competitive. Furthermore, the pull latency, leveraging the optimistic fallback to the middleman, is only marginally slower than a 'git pull', confirming the system's viability for frequent, read-heavy collaborative tasks.

**System overhead and bottleneck analysis.** While the user experiences a fast push, the system performs the blockchain registration in the background to ensure long-term security and auditability. Fig 4 visualizes the full system workload, including this asynchronous blockchain component.

The data clearly identifies the blockchain confirmation time as the single largest component of the total system workload, representing a "decentralization tax" of approximately 12-16 seconds per transaction. By handling this process asynchronously, our architecture provides the user with the speed of a centralized system while still gaining the security benefits of an immutable on-chain record. Furthermore, the on-chain gas cost for registering a repository was consistently measured at **206,886 gas**, confirming that our design remains cost-effective by storing only minimal data on-chain, regardless of file size.

**Validation of the optimistic retrieval protocol.** The effectiveness of our design hinges on our retrieval protocol, which prioritizes authority while mitigating latency. We validated

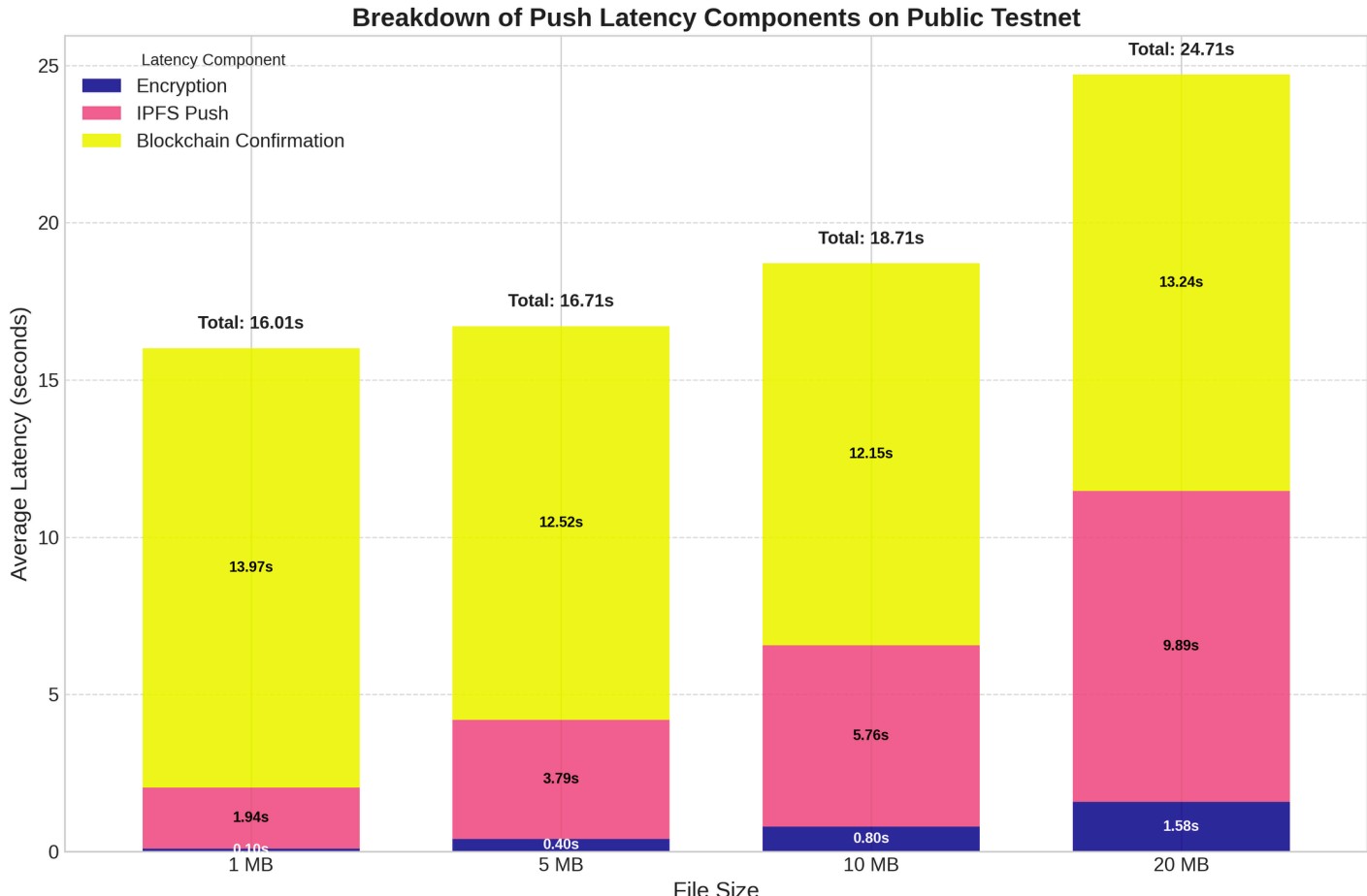

**Fig 4. Breakdown of total system workload during a push operation:** The `Pure Blockchain Latency` represents a large, constant overhead, confirming it as the primary system bottleneck that our asynchronous design successfully abstracts away from the user's workflow.

this by comparing system performance in an idealized local environment against the public testnet, as shown in Fig 5.

The local testbed results establish a performance baseline for the system's core logic, free from network latency. The public testnet results confirm that our system successfully navigates real-world network delays. Crucially, our experiments demonstrated the success of the retrieval logic in both pre- and post-confirmation scenarios. When retrieval was attempted before blockchain confirmation, the authoritative on-chain call correctly failed, triggering the optimistic fallback to the middleman for the necessary key share. This allowed for immediate file access, proving the system's ability to bypass user-facing latency. Conversely, when retrieval was attempted after confirmation, the system successfully retrieved the share from the authoritative on-chain source first, never needing to contact the less-trusted middleman. This experiment empirically proves that our hybrid design effectively and intelligently decouples the user experience from blockchain finality, solving a critical usability challenge for decentralized applications.

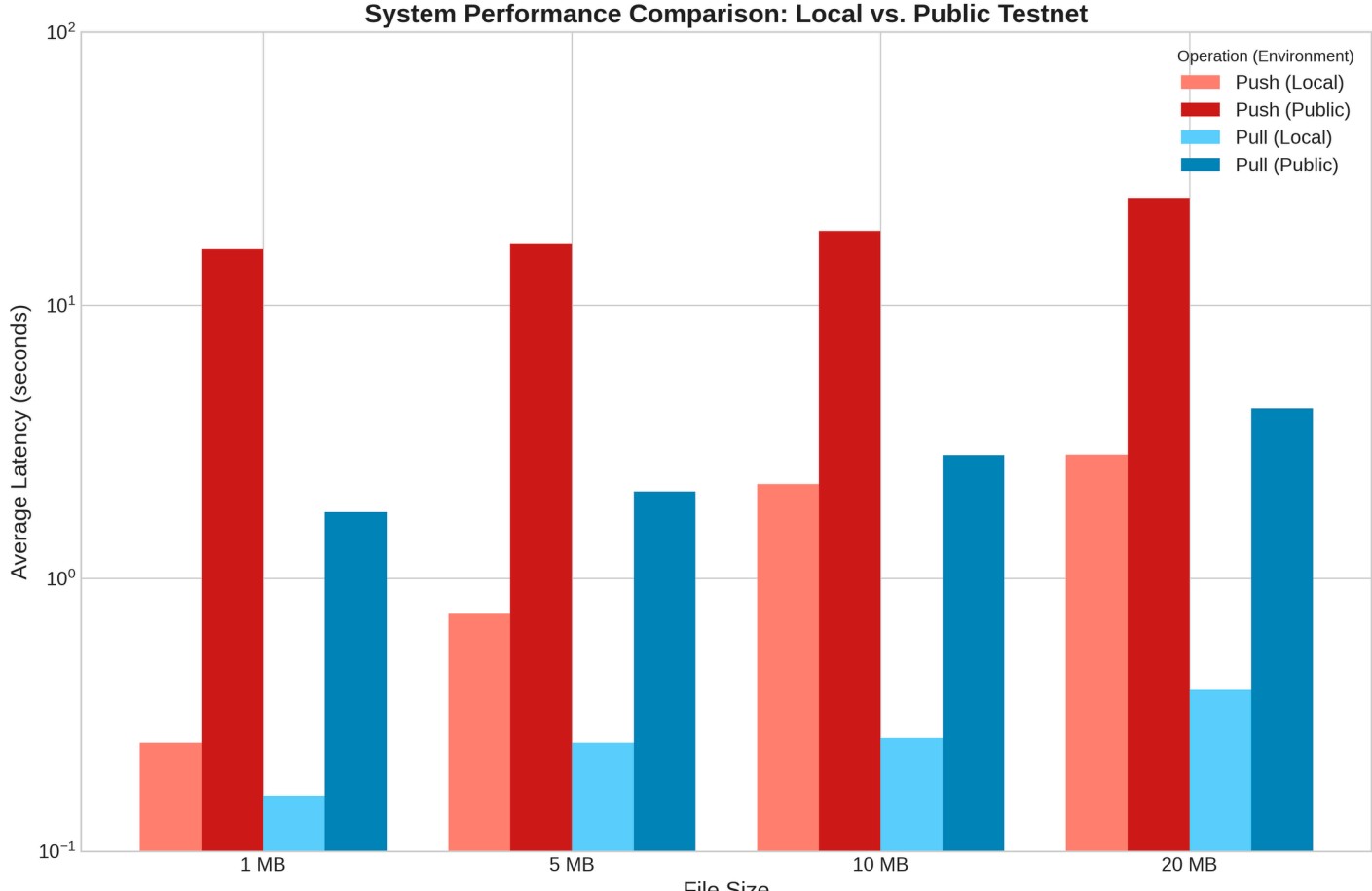

**Fig 5. System latency comparison: Local vs. public testnet environments.** This chart quantifies the performance overhead of public networks. Minimal local latency (lighter bars) reflects efficient core logic, while higher public latency (darker bars) is attributable to real-world network and consensus delays.

## Security and trust model

Our system's security model is designed to minimize trust and eliminate single points of failure, primarily through the integration of Shamir's Secret Sharing (SSS).

By splitting the encryption key into a (2,3)-threshold scheme, we render the centralized middleman component "cryptographically powerless." A malicious or compromised middleman only possesses one of the three shares, which, by the properties of SSS, reveals no information about the original key. An attacker would need to compromise the middleman and either the owner or the blockchain simultaneously to reconstruct the key, a significantly higher security barrier than traditional systems.

Furthermore, the authoritative copy of a key share is stored immutably on the Ethereum blockchain. This ensures that even if the temporary middleman fails or its data is lost, the repository remains permanently recoverable via the on-chain record. This design directly addresses the key retention and deletion concerns; deletion from the middleman is not a critical security requirement, as its share is both temporary for performance and insufficient for an attack. The blockchain serves as the ultimate source of truth and disaster recovery.

## Developer workflow and usability

While our prototype utilizes a web-based graphical user interface, the underlying cryptographic and network operations are self-contained, making them suitable for integration into automated development pipelines. A future command-line interface (CLI) could wrap the core `push` and `pull` functions, allowing them to be scripted. This would enable the integration of our decentralized version control system into Continuous Integration/Continuous Deployment (CI/CD) workflows. For instance, a CI pipeline could be configured to automatically pull the latest version from a specific `ipfsHash`, run automated tests, and, upon success, use a securely stored owner's key share to programmatically push the new build artifacts as a new version, creating a fully decentralized and verifiable build and release process.

## Conclusion

This work successfully designed, implemented, and evaluated a secure and efficient decentralized version control system that overcomes the critical latency issues plaguing many blockchain-based solutions. By integrating an authoritative-first, optimistic-fallback protocol with a Shamir's Secret Sharing security model, our hybrid architecture delivers on the promise of decentralization—providing immutable ownership records and enhanced resilience—without sacrificing the performance necessary for real-time collaboration.

Our comprehensive experiments on the public Sepolia testnet provide two key findings. First, by architecturally separating the user's interactive workflow from on-chain settlement, the user-perceived push latency of our system is highly competitive and outperforms a centralized Git/GitHub baseline for repositories up to 10MB. Second, the pull latency, which leverages the optimistic fallback mechanism, is only marginally slower than its centralized counterpart, confirming the system's viability for frequent, read-heavy developer tasks. We have empirically demonstrated that the primary system overhead is the asynchronous blockchain confirmation time, a "decentralization tax" that our architecture successfully abstracts away from the user's critical path.

Future work can build upon this validated foundation. While our system excels at core push and pull operations, further research could explore the implementation of more complex Git-native features like branching and merging within this decentralized paradigm. Additionally, optimizing the middleman component by transitioning it to a Decentralized Autonomous Organization (DAO) for coordination could further enhance the system's resilience. Such a DAO could, for example, manage a treasury of funds to pay for IPFS pinning services or automatically prune expired key shares from a network of federated middleman nodes. Governance could be managed via tokens, where stakeholders could vote on protocol upgrades or the inclusion of new, trusted middleware providers. This would further align the system with a fully trustless ethos. Ultimately, our study proves that a thoughtfully designed hybrid system can strike an effective balance between the speed of centralized services and the robust security of decentralized technologies.

## Supporting information

**S1 Appendix. Detailed experimental data.** This appendix contains the detailed, trial-by-trial results and summary tables from the performance evaluation for the public testnet trials. (PDF)

## Author contributions

**Conceptualization:** Md. Rafid Haque, Md. Tahmid Islam, A.B.M. Ashikur Rahman.

**Formal analysis:** Md. Tahmid Islam.

**Investigation:** Md. Rafid Haque, Sakibul Islam Munna.

**Methodology:** Md. Rafid Haque, Sakibul Islam Munna, Md. Tahmid Islam.

**Software:** Md. Rafid Haque.

**Supervision:** Sabbir Ahmed, Md Mehedi Hassan Onik, A.B.M. Ashikur Rahman.

**Visualization:** Md. Rafid Haque, Sakibul Islam Munna.

**Writing – original draft:** Md. Rafid Haque, Sakibul Islam Munna.

**Writing – review & editing:** Sabbir Ahmed, Md Mehedi Hassan Onik, A.B.M. Ashikur Rahman.

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
