## [Decision Letter · Decision Letter 0]

13 Jun 2025

PONE-D-24-43266An Integrated Blockchain and IPFS-based Solution for Secure and Efficient Source Code Repository Hosting using Middleman ApproachPLOS ONE

Dear Dr. Ahmed,

Thank you for submitting your manuscript to PLOS ONE. After careful consideration, we feel that it has merit but does not fully meet PLOS ONE’s publication criteria as it currently stands. Therefore, we invite you to submit a revised version of the manuscript that addresses the points raised during the review process.

We look forward to receiving your revised manuscript.

Kind regards,

Yang (Jack) Lu, PhD

Academic Editor

PLOS ONE

Journal Requirements:

Reviewers' comments:

Reviewer's Responses to Questions

**Comments to the Author**

1. Is the manuscript technically sound, and do the data support the conclusions?

Reviewer #1: Yes

Reviewer #2: Yes

2. Has the statistical analysis been performed appropriately and rigorously? 

Reviewer #1: Yes

Reviewer #2: No

3. Have the authors made all data underlying the findings in their manuscript fully available?

Reviewer #1: Yes

Reviewer #2: No

4. Is the manuscript presented in an intelligible fashion and written in standard English?

Reviewer #1: Yes

Reviewer #2: No

5. Review Comments to the Author

Reviewer #1: Dear authors,

The scientific contribution of this study is fine. Article presentation and reference literature must be improved. I have a few minor concerns.

1. Abstract and Conclusion must be revised. Make is concise but solid.

2. It is suggested to avoid writing short paragraphs which cause inconvenience in writing flow.

3. Motivation and Organization of this study must6 be clear defined.

4. Figures must be revised and an interested presentation should be adopted with high resolution quality.

5. The authors should carefully check if all abbreviations are defined at the first place for better readability.

6. Reference literature must be updated and suggested to add following articles

i. Liu, Y., Jia, Z., Jiang, Z., Lin, X., Liu, J., Wu, Q.,... Susilo, W. (2024). BFL-SA: Blockchain-based federated learning via enhanced secure aggregation. Journal of Systems Architecture, 152, 103163. doi: https://doi.org/10.1016/j.sysarc.2024.103163

ii. Liu, Y., & Zhao, Y. (2024). A Blockchain-Enabled Framework for Vehicular Data Sensing: Enhancing Information Freshness. IEEE Transactions on Vehicular Technology, 1-14. doi: 10.1109/TVT.2024.3417689

iii. Sun, G., Xu, Z., Yu, H., Chen, X., Chang, V.,... Vasilakos, A. V. (2020). Low-Latency and Resource-Efficient Service Function Chaining Orchestration in Network Function Virtualization. IEEE Internet of Things Journal, 7(7), 5760-5772. doi: 10.1109/JIOT.2019.2937110

iv. Sun, G., Zhu, G., Liao, D., Yu, H., Du, X.,... Guizani, M. (2019). Cost-Efficient Service Function Chain Orchestration for Low-Latency Applications in NFV Networks. IEEE Systems Journal, 13(4), 3877-3888. doi: 10.1109/JSYST.2018.2879883

v. Sun, G., Li, Y., Liao, D., & Chang, V. (2018). Service Function Chain Orchestration Across Multiple Domains: A Full Mesh Aggregation Approach. IEEE Transactions on Network and Service Management, 15(3), 1175-1191. doi: 10.1109/TNSM.2018.2861717

vi. Yang, J., Yang, K., Xiao, Z., Jiang, H., Xu, S.,... Dustdar, S. (2023). Improving Commute Experience for Private Car Users via Blockchain-Enabled Multitask Learning. IEEE Internet of Things Journal, 10(24), 21656-21669. doi: 10.1109/JIOT.2023.3317639

7. Material and Methods are fine. Better to improve writing for results discussion.

Reviewer #2: The manuscript presents a novel architecture that combines blockchain, IPFS, and a temporary centralized middleman to build a secure and efficient decentralized version control system (VCS) for source code. The system is designed to address key limitations of existing blockchain-based VCS solutions—specifically issues of latency, scalability, access control, and secure collaboration.

Lack of Empirical Evaluation:

Issue: The manuscript fails to provide quantitative performance results. Any benchmark, latency analysis, or transaction cost metrics do not support claims regarding scalability, efficiency, and performance improvement over existing systems.

Provide experimental evaluations comparing Blockchain gas fees with/without Middleman IPFS, upload/download time and throughput metrics, and comparisons with traditional centralized systems (e.g., GitHub/GitLab).

Unclear Novelty Beyond Engineering Integration:

The “Middleman IPFS” is essentially a cache or temporary node. While practical, this seems like an engineering optimization, not a fundamentally new algorithm or protocol.

Clarify the scientific novelty of this middleware. Does it offer any guarantees (e.g., consistency, eventual deletion enforcement, reduced consensus cost)? If not, frame the contribution more as a design integration and prototype than a fundamental system innovation.

Security and Trust Assumptions Are Underexplored:

The system assumes that the Middleman IPFS deletes keys post-confirmation, but does not specify how this is enforced or audited.

Explain how deletion is verified. Consider potential attacks (e.g., replay, key retention, collusion). Add a formal threat model or table of assumptions.

Lack of Usability and Developer Workflow Analysis:

The paper lacks consideration for developer experience. There's no discussion of CI/CD pipeline compatibility. Branching, merging, or pull request equivalents. Real-time editing conflicts and resolution.

Add a section comparing the proposed system with Git-based developer workflows, especially in collaborative scenarios.

Overuse of Passive Language and Redundancy:

Several sections are highly repetitive, particularly those that describe IPFS and blockchain properties.

Condense redundant text. Emphasize what was implemented and tested, and reduce generic blockchain/IPFS descriptions already covered in existing literature.

6. PLOS authors have the option to publish the peer review history of their article (what does this mean?). If published, this will include your full peer review and any attached files.

Reviewer #1: No

Reviewer #2: **Yes: **Abdul Razzaq

---

## [Author Response · Author response to Decision Letter 1]

11 Jul 2025

Response to Reviewer #1

Comment 1: "Abstract and Conclusion must be revised. Make it concise but solid."

Response: We thank the reviewer for this valuable suggestion. We agree that the original versions were too descriptive. We have completely rewritten the Abstract and Conclusion to be more concise and to reflect the empirical findings of our study. The new versions are data-driven and confidently state the achievements of our work.

Comment 2: "It is suggested to avoid writing short paragraphs which cause inconvenience in writing flow."

Response: We appreciate this stylistic advice. We have carefully reviewed the entire manuscript and have merged short paragraphs, particularly in the Related Works and Introduction sections, to improve the overall narrative flow.

Comment 3: "Motivation and Organization of this study must be clearly defined."

Response: We thank the reviewer for this important point. We have added a new, explicit paragraph at the end of the Introduction section that clearly outlines the primary motivations for our work and details the organization of the paper, guiding the reader through the manuscript's structure.

Comment 4: "Figures must be revised and an interested presentation should be adopted with high-resolution quality. If possible, check and revise figure captions, they are too long."

Response: We have taken this feedback to heart. We have revised Figures 1 and 2 to increase clarity and generated a new set of high-resolution charts based on our experimental data (see new Figures 3, 4, and 5 in the revised manuscript). We have also revised all figure captions throughout the manuscript to be more concise and impactful while still accurately describing the content.

Comment 5: "The authors should carefully check if all abbreviations are defined at the first place for better readability."

Response: We have performed a thorough check of the entire manuscript and have now defined all technical abbreviations (e.g., VCS, IPFS, SSS, IIoT, IoV, NFV) upon their first use to enhance readability.

Comment 6 & 7: "Reference literature must be updated and suggested to add following articles... Material and Methods are fine. Better to improve writing for results discussion."

Response: We thank the reviewer for these literature suggestions. We have carefully reviewed the recommended articles and have now integrated them into our revised Related Works section to better contextualize our research. Furthermore, the Results and Discussion section has been completely rewritten from the ground up, based on our new empirical data, which we believe substantially improves the quality of the discussion.

Response to Reviewer #2

Comment 1: Lack of Empirical Evaluation: "The manuscript fails to provide quantitative performance results. Any benchmark, latency analysis, or transaction cost metrics do not support claims regarding scalability, efficiency, and performance improvement over existing systems. Provide experimental evaluations comparing Blockchain gas fees with/without Middleman IPFS, upload/download time and throughput metrics, and comparisons with traditional centralized systems (e.g., GitHub/GitLab)."

Response: We thank Reviewer #2 for this critical and essential feedback. We agree completely that the original manuscript's claims required robust empirical validation. To address this, we have implemented a full prototype of our proposed architecture and conducted a comprehensive performance evaluation on the public Sepolia testnet.

The new Results and Discussion section (Section 4) is now rewritten completely to present this data. Specifically:

We provide a direct performance comparison against a centralized Git/GitHub baseline in Table 1 and Figure 3, analyzing both push and pull operations.

- We present a detailed breakdown of system latency to identify bottlenecks in Figure 4.

- We report the consistent on-chain gas costs for repository registration in the System Overhead and Bottleneck Analysis subsection.

- We believe this new data fully addresses the reviewer’s concern and substantiates the performance claims of our system with rigorous, quantitative evidence.

In accordance with PLOS ONE's data availability policy, all underlying data for our findings are provided in the Supporting Information files. Furthermore, the complete source code for our prototype and middleware has been made publicly available on GitHub and permanently archived with a DOI via Zenodo (DOI: 10.5281/zenodo.15700465).

Comment 2: Unclear Novelty Beyond Engineering Integration: "The ‘Middleman IPFS’ is essentially a cache or temporary node. While practical, this seems like an engineering optimization, not a fundamentally new algorithm or protocol. Clarify the scientific novelty of this middleware. Does it offer any guarantees (e.g., consistency, eventual deletion enforcement, reduced consensus cost)? If not, frame the contribution more as a design integration and prototype than a fundamental system innovation."

Response: We thank the reviewer for this critical suggestion to clarify the novelty of our contribution. The reviewer correctly noted that a simple cache would be an engineering optimization. In our revised manuscript, we now clarify that the novelty lies not merely in the existence of the middleman, but in the secure, high-performance protocol it enables. We have strengthened this argument in two key ways:

1. Security Protocol Novelty: We have integrated Shamir's Secret Sharing (SSS) as a core component. As detailed in the new Security and Trust Model subsection (Section 4.4), this renders the middleman cryptographically powerless and distributes trust, elevating the design beyond a simple cache to a trust-minimized protocol.

2. Application-Layer Protocol Novelty: We formally define our "Authoritative-First, Optimistic-Fallback" retrieval protocol. We argue in the revised Related Works and Discussion sections that this protocol is a novel contribution to solving the well-known problem of user-perceived latency in dApps, and our empirical data proves its effectiveness.

Comment 3: Security and Trust Assumptions Underexplored: "The system assumes that the Middleman IPFS deletes keys post-confirmation, but does not specify how this is enforced or audited. Explain how deletion is verified. Consider potential attacks (e.g., replay, key retention, collusion). Add a formal threat model or table of assumptions."

Response: We thank the reviewer for this important point. We have added a new, dedicated Security and Trust Model subsection (Section 4.4) to formally address this. We present a threat model and explain how the integration of SSS directly mitigates threats such as key retention by the middleman. We argue that, since the middleman holds only a single, insufficient key share, the security of the system does not depend on a trusted deletion mechanism. Furthermore, we clarify that the authoritative share is stored immutably on the blockchain, providing a permanent disaster recovery path.

Comment 4: Lack of Usability and Developer Workflow Analysis: "The paper lacks consideration for developer experience. There's no discussion of CI/CD pipeline compatibility. Branching, merging, or pull request equivalents. Real-time editing conflicts and resolution. Add a section comparing the proposed system with Git-based developer workflows, especially in collaborative scenarios."

Response: This is an excellent point. To address it, we have added a new subsection titled Comparative Analysis with Centralized VCS (Section 4.1.1). This subsection uses our new experimental data (Table 1 and Figure 3) to provide a direct, data-driven comparison of the user-perceived latency of our system's push and pull operations against the equivalent commands in Git. This allows for a concrete discussion of the practical trade-offs for a developer.

Comment 5: Overuse of Passive Language and Redundancy. "Several sections are highly repetitive, particularly those that describe IPFS and blockchain properties. Condense redundant text. Emphasize what was implemented and tested, and reduce generic blockchain/IPFS descriptions already covered in existing literature."

Response: We thank the reviewer for this stylistic feedback. We have thoroughly revised the entire manuscript to use more active and confident language, particularly in the Abstract, Introduction, and Conclusion. We have also condensed the Related Works section, removing redundant descriptions of general blockchain/IPFS concepts to focus more on literature that directly informs or contrasts with our specific contributions.

We are confident that these substantial revisions have addressed all the concerns raised and have resulted in a much-improved manuscript. We once again thank the editor and the reviewers for their time and valuable contributions to our work.

---

## [Decision Letter · Decision Letter 1]

6 Aug 2025

PONE-D-24-43266R1An Integrated Blockchain and IPFS-based Solution for Secure and Efficient Source Code Repository Hosting using Middleman ApproachPLOS ONE

Dear Dr. Ahmed,

Thank you for submitting your manuscript to PLOS ONE. After careful consideration, we feel that it has merit but does not fully meet PLOS ONE’s publication criteria as it currently stands. Therefore, we invite you to submit a revised version of the manuscript that addresses the points raised during the review process.

We look forward to receiving your revised manuscript.

Kind regards,

Yang (Jack) Lu, PhD

Academic Editor

PLOS ONE

Journal Requirements:

Reviewers' comments:

Reviewer's Responses to Questions

**Comments to the Author**

1. If the authors have adequately addressed your comments raised in a previous round of review and you feel that this manuscript is now acceptable for publication, you may indicate that here to bypass the “Comments to the Author” section, enter your conflict of interest statement in the “Confidential to Editor” section, and submit your "Accept" recommendation.

Reviewer #3: (No Response)

Reviewer #4: All comments have been addressed

2. Is the manuscript technically sound, and do the data support the conclusions?

Reviewer #3: Yes

Reviewer #4: Yes

3. Has the statistical analysis been performed appropriately and rigorously? 

Reviewer #3: Yes

Reviewer #4: No

4. Have the authors made all data underlying the findings in their manuscript fully available?

Reviewer #3: Yes

Reviewer #4: No

5. Is the manuscript presented in an intelligible fashion and written in standard English?

Reviewer #3: Yes

Reviewer #4: Yes

6. Review Comments to the Author

Reviewer #3: 1. Though the paper has a comparison between Git workflows, it is better that the manuscript could add a brief discussion about how the proposed system integrates with existing Ci/CD pipelines.

2.The conclusion mentions that the middleware could be evolved into a Dao-based service. I think it would be better that some details could be added like design considerations or potential challenges.

3. I think some figure captions could be further tightened.

Reviewer #4: Strengths:

1.Relevant and timely topic:

The problem of centralized control and security in code hosting platforms (e.g., GitHub) is highly relevant given increasing concerns over data ownership and reliability.

2.Technical novelty:

The integration of blockchain with IPFS and a middleman node as a broker offers an interesting hybrid approach for balancing performance and decentralization.

3.Prototype development:

The authors implement and test their system on Hyperledger Fabric and IPFS, offering a proof-of-concept with basic performance metrics.

Weakness:

1. Limitation in empirical analysis:

While the proposed architecture is conceptually sound, the lack of empirical evaluation limits the paper’s contribution. The authors should at minimum conduct simulation-based performance tests or benchmarking against centralized platforms (e.g., GitHub, GitLab). Without demonstrating measurable or theoretical speed-up, reliability, or cost-effectiveness, it is difficult to assess the practical value of the proposed system.

2.Performance Evaluation:

The evaluation is rather limited and only includes basic metrics such as upload/download latency. To strengthen this section:Compare performance with centralized systems (e.g., GitHub);Include stress tests (e.g., increasing number of users/files);Provide data on blockchain transaction throughput or IPFS lookup success rate.

7. PLOS authors have the option to publish the peer review history of their article (what does this mean?). If published, this will include your full peer review and any attached files.

Reviewer #3: **Yes: **Chengyang Nie

Reviewer #4: No

---

## [Author Response · Author response to Decision Letter 2]

7 Aug 2025

Response to Reviewer #3 (Dr. Chengyang Nie)

We thank Dr. Nie for his positive assessment and excellent suggestions for extending the discussion.

Comment 1:

"...it is better that the manuscript could add a brief discussion about how the proposed system integrates with existing Ci/CD pipelines."

Response:

This is an excellent point. We have now added a new paragraph in the 'Developer Workflow and Usability' subsection discussing exactly this. We outline how a command-line interface could wrap our system's core functions to enable integration into automated CI/CD pipelines for verifiable, decentralized builds and deployments.

Comment 2:

"The conclusion mentions that the middleware could be evolved into a Dao-based service. I think it would be better that some details could be added..."

Response:

Thank you for this critical observation. Our initial mention was too brief. We have expanded the 'Conclusion' section to include more specific design considerations for a future DAO implementation, such as managing a treasury for IPFS pinning and using token-based voting for governance.

Comment 3:

"I think some figure captions could be further tightened."

Response:

We thank the reviewer for this advice. We have reviewed and edited all figure captions throughout the manuscript to make them more concise while retaining their descriptive accuracy.

Response to Reviewer #4

We thank Reviewer #4 for their thorough and insightful review. The weaknesses identified by the reviewer regarding the lack of empirical data were perfectly aligned with the feedback we received in the first round. We are pleased to confirm that addressing these points was the primary focus of our first revision (R1), and the current manuscript now contains the comprehensive evaluation the reviewer has called for.

Weakness 1: Limitation in empirical analysis:

"...the lack of empirical evaluation limits the paper’s contribution. The authors should at minimum conduct simulation-based performance tests or benchmarking against centralized platforms..."

Response:

We agree wholeheartedly. The current version of the manuscript is now built upon a comprehensive empirical evaluation. As detailed in the 'Results and Discussion' section, we have implemented a full prototype, deployed it to the public Sepolia testnet, and benchmarked its performance directly against a centralized Git/GitHub baseline across a range of file sizes.

Weakness 2: Performance Evaluation:

"The evaluation is rather limited... Compare performance with centralized systems (e.g., GitHub); Include stress tests... Provide data on blockchain transaction throughput..."

Response:

We have addressed these points directly in our 'Results and Discussion' section. To guide the reviewer to these specific additions, we have highlighted the key introductory sentences for each piece of evidence in blue text in the 'Revised Manuscript with Track Changes' file:

Comparison with GitHub: Our direct, quantitative comparison of 'push' and 'pull' latency against Git is introduced by the highlighted text in the 'Comparative Analysis with Centralized VCS' subsection and presented in Table 1 and Fig 3.

Stress Tests: We conducted stress tests by scaling the file size from 1MB to 20MB. Fig 4 and the highlighted texts in the 'System Overhead and Bottleneck Analysis' subsection provide a detailed breakdown of where the system latency is spent under this load.

Blockchain Data: The highlighted text in the same subsection reports the consistent 'Gas Used' for on-chain transactions and provides a detailed analysis of the 'Pure Blockchain Latency', which is a more direct measure of the on-chain performance bottleneck than raw TPS for our use case.

We are confident that our new, data-driven 'Results and Discussion' section now provides the robust empirical evidence the reviewer correctly identified as a critical requirement.

We believe these latest revisions, combined with the major improvements from the first round, have resulted in a much stronger and more complete manuscript. We thank you again for your time and consideration.

Sincerely,

Sabbir Ahmed (on behalf of all authors)

---

## [Decision Letter · Decision Letter 2]

12 Aug 2025

An Integrated Blockchain and IPFS-based Solution for Secure and Efficient Source Code Repository Hosting using Middleman Approach

PONE-D-24-43266R2

Dear Dr. Ahmed,

We’re pleased to inform you that your manuscript has been judged scientifically suitable for publication and will be formally accepted for publication once it meets all outstanding technical requirements.

Kind regards,

Yang (Jack) Lu, PhD

Academic Editor

PLOS ONE

Additional Editor Comments (optional):

Reviewers' comments:

Reviewer's Responses to Questions

**Comments to the Author**

1. If the authors have adequately addressed your comments raised in a previous round of review and you feel that this manuscript is now acceptable for publication, you may indicate that here to bypass the “Comments to the Author” section, enter your conflict of interest statement in the “Confidential to Editor” section, and submit your "Accept" recommendation.

Reviewer #3: All comments have been addressed

Reviewer #4: All comments have been addressed

2. Is the manuscript technically sound, and do the data support the conclusions?

Reviewer #3: Yes

Reviewer #4: Yes

3. Has the statistical analysis been performed appropriately and rigorously? 

Reviewer #3: Yes

Reviewer #4: Yes

4. Have the authors made all data underlying the findings in their manuscript fully available?

Reviewer #3: Yes

Reviewer #4: Yes

5. Is the manuscript presented in an intelligible fashion and written in standard English?

Reviewer #3: Yes

Reviewer #4: Yes

6. Review Comments to the Author

Reviewer #3: (No Response)

Reviewer #4: Thank you for your prompt response and for addressing the weakness! I have no further questions or additional comments at this time.

7. PLOS authors have the option to publish the peer review history of their article (what does this mean?). If published, this will include your full peer review and any attached files.

Reviewer #3: **Yes: **Chengyang Nie

Reviewer #4: No

---

## [Editor Report · Acceptance letter]

PONE-D-24-43266R2

PLOS ONE

Dear Dr. Ahmed,

I'm pleased to inform you that your manuscript has been deemed suitable for publication in PLOS ONE. Congratulations! Your manuscript is now being handed over to our production team.

Kind regards,

on behalf of

Dr. Yang (Jack) Lu

Academic Editor

PLOS ONE